# Breastfeeding and Allergic Diseases: What’s New?

**DOI:** 10.3390/children8050330

**Published:** 2021-04-24

**Authors:** Giulia Nuzzi, Maria Elisa Di Cicco, Diego Giampietro Peroni

**Affiliations:** Section of Pediatrics, Department of Clinical and Experimental Medicine, University of Pisa, 56126 Pisa, Italy; giulianuzzi92@gmail.com (G.N.); maria.dicicco@unipi.it (M.E.D.C.)

**Keywords:** breastfeeding, allergic diseases, asthma, atopic dermatitis, food allergy, primary prevention, human milk

## Abstract

Asthma and other allergic disorders, such as atopic dermatitis and food allergies, are common chronic health problems in childhood. The rapid rise in the prevalence of these conditions registered over the last few decades has stressed the need to identify the modifiable risk factors associated with the development of these diseases. Breast milk, recognized as the gold standard for healthy growth and development of the newborn, is one of the major factors associated with a lower incidence of allergic and infectious diseases in childhood and young adulthood. Although the underlying mechanisms for these effects are not well understood, breastfeeding leads to immune system maturation. In this narrative review, we summarize existing evidence on breastfeeding and human milk composition in relation to allergic disease prevention or development.

## 1. Introduction

Over 300 million people worldwide suffer from asthma, while allergic rhinitis and other allergic diseases, such as atopic dermatitis (AD) and food allergies, affect between 10 and 30% of the global population [1,2]. The rapid rise in the prevalence of these conditions registered over the last few decades, and their negative impacts on health-related quality of life, has stressed the need to identify the environmental and modifiable risk factors associated with the development of these diseases. Explanations for this rapid increase in allergies are not entirely clear, though the “hygiene hypothesis” remains the most widely cited theory [3]. This hypothesis explains the increase in allergies as a mutually counter-regulatory interaction between the immune response to infection and that associated with allergy [4], although other factors have been brought forward, such as important changes in dietary preferences over the last few years.

In recent years, research has focused on both prenatal and postnatal modifiable factors for allergic disease prevention, as these factors can affect the immune system in an important phase of its development [5]. Among these, breastfeeding seems to be the most relevant postnatal factor that drives the immune system development of the newborn [6]. Increasing evidence shows that breastfeeding plays a central role in tolerogenic immune responses during the first years of life [7]. Human milk has been shown to protect against early respiratory and other infections, due to its nutritional composition and content in non-nutritive bioactive factors i.e., immunoglobulins, vitamin A, and transforming growth-factors. These factors promote the gut mucosal barrier integrity and protect against severe conditions, such as necrotizing enterocolitis, diabetes, obesity, infections, and allergic diseases, as well as reduce the risk of health-related problems later in life [8,9]. Indeed, growing evidence supports the hypothesis that colonization by a healthy microbiome during the first 1000 days after conception can affect the immune system development and the predisposition to immune-mediated diseases later in life, including asthma [10]. For these reasons, exclusive breastfeeding for at least the first 6 months of life continued up to 2 years of age is considered the gold standard for infant feeding. Because human milk is uniquely suited to the human infant, both in its nutritional content and in its bioactivity, breastfeeding promotes the healthy development of the human being, as well as having numerous health benefits for the mother [11,12]. It is known that breastfeeding increases levels of oxytocin [13], resulting in less postpartum bleeding and more rapid uterine involution [14]. Recent research demonstrates that lactating women have an earlier return to prepregnant weight [15], delayed resumption of ovulation with increased child spacing, improved bone remineralization postpartum [16], and reduced risk of ovarian cancer and premenopausal breast cancer [17,18]. In addition to individual health benefits, breastfeeding provides significant social and economic benefits to the nation, including reduced health care costs and reduced employee absenteeism for care attributable to child illness. As a matter of fact, the significantly lower incidence of illness in the breastfed infant allows the parents more time for attention to siblings and other family duties and reduces parental absence from work and lost income [11].

Despite all of these benefits, the protection afforded by human milk against the development of childhood allergic disease has been the subject of controversy in the literature in the last few years. Indeed, although breastfeeding is considered protective against respiratory tract infections [19,20], this protection has not been demonstrated for asthma and allergic diseases in all studies. Issues related to study design and analytical methods greatly complicate the interpretation of studies.

## 2. Composition of Human Milk

From an evolutionary and nutritional standpoint, human milk is recognized as the gold standard of nourishment for human infants in the first months of life, because it is a species-specific food, adapted to provide what is needed for optimal growth and survival to the offspring, with a unique composition designed by nature to better respond to the biological and psychological needs of the newborn. Indeed, in children, breastfeeding has been associated with improved cognitive performance and socio-affective responses [21]. Improved cognitive performance is probably linked to the fatty acids contained in human milk and their beneficial effect on brain development, while heightened socio-affective responding seen in breastfed children is likely connected to the stimulation of the oxytocin system, known to have a key role in promoting positive affects and approach behaviors [22]. However, it should be emphasized that breastfeeding is associated with higher socioeconomic status and the education status of the mother, as well as less smoking and obesity. These factors alone could affect the infant’s development, leading to a benefit that could be attributed to breastfeeding itself, without concrete evidence of the cause of this intervention. Breast milk composition is dynamic and varies within a feeding, diurnally, over lactation, and between mothers and populations [23]. The first fluid produced after delivery is colostrum. Colostrum, produced in low quantities in the first few days postpartum, is rich in immunologic components such as secretory IgA, lactoferrin, leukocytes, as well as developmental factors such as epidermal growth factor [23,24]. After colostrum, the composition of human milk undergoes alterations throughout the lactation period which correspond to the nutritional needs of the newborn in the different stages of its growth. Transitional milk has some colostrum-like characteristics but represents a period of increased milk production to support the nutritional needs of the rapidly growing infant; it typically occurs from five days to two weeks postpartum. By four to six weeks postpartum, human milk is considered fully mature.

### 2.1. Macro- and Micronutrients in Human Milk

After delivery, when the infant host defenses are vulnerable, human milk provides protection through transference of antimicrobial and anti-inflammatory compounds, while also stimulating the immune system maturation. In addition, breast milk contains prebiotic compounds (i.e., oligosaccharides, non-digestible nutrients that benefit the host by selectively stimulating the growth or activity of a bacterial species present in the intestinal bacterial flora), as well as its own microbiota, both supplying the initial gut colonizers and supporting the microbiota development in the infant gut [25]. Indeed, term infants born vaginally and breastfed exclusively seem to have the most “beneficial” gut microbiome, with the highest concentration of *Bifidobacteria* and lowest numbers of *Clostridium difficile* and *Escherichia coli* [26,27].

Primary components of breast milk are carbohydrates and fatty acids, which play the main nutritional role of breast milk. Fat is the most variable macronutrient of human milk, containing high levels of palmitic and oleic acids [28]. The principal sugar of human milk is the disaccharide lactose, while other significant carbohydrates of human milk are the oligosaccharides (HMOs), which comprise approximately 1 g/dL in human milk, depending on the stage of lactation and maternal genetic factors. Though non-nutritive to the infant, HMOs constitute a remarkable quantity of human milk, recognized as pathogen-binding inhibitors that function as soluble receptors for pathogens that have an affinity for binding to oligosaccharide receptors expressed on the newborn’s intestinal surface [23]. Although human milk provides the ideal food for infant nourishment, it doesn’t meet the Recommended Dietary Allowances (RDA) for all vitamins because many micronutrients vary in breast milk depending on maternal diet and body stores [29]. The most glaring discrepancy between intake and RDA is for vitamin D: although infants may synthesize it from sunlight exposure, American Academy of Pediatrics (AAP) recommendations target postnatal vitamin D supplementation of breastfed infants. AAP also recommend an injection of vitamin K in the first hours after birth to avoid the hemorrhagic disease of the newborn, as vitamin K is extremely low in human milk [23].

### 2.2. Bioactive Compounds in Human Milk

Human milk contains a wide variety of bioactive factors that stimulate host defenses development in the newborn, including enzymes, hormones, growth factors, lactoferrin, cytokines, immunological agents, and other immunomodulating molecules. The most abundant proteins are casein, α-lactalbumin, lactoferrin, secretory immunoglobulin A (sIgA), lysozyme, and serum albumin [30]. Lactoferrin is a glycoprotein that binds iron, limiting the availability of iron to pathogens and preventing them from binding to the intestinal wall. Cytokines, antibodies and lysozyme are all mature immune system components. Antibodies, particularly secretory IgA, pass from mother to child trough breast milk and, as well as lactoferrin, prevent pathogens from binding to intestinal wall while lysozyme directly attacks bacterial cells and cytokines modulate intestinal inflammation. Cytokines are small soluble glycoproteins which act by binding to specific cellular receptors and modulating immune system development and function [31]. Transforming growth factor-β (TGF-β) is a human milk cytokine, that may influence the development and maturation of the mucosal immune system of the infant: evidence suggests that TGF-β may be a key immunoregulatory factor for the establishment of this response, by promoting IgA production as well as induction of oral tolerance [32,33]. Since the immune system takes time to mature and evolve, the infant initially relies on the innate intestinal immune system, which is helped to develop by the bioactive compounds contained in human milk.

## 3. Association between Breastfeeding and Allergic Diseases

Asthma and other allergic conditions result from a complex interaction of genetic and environmental factors, such as early childhood feeding, exposure to passive cigarette smoking, and domestic allergens (i.e., dust mites or animal furfur), which play an important but complex role in the risk of atopy development. Early infancy, particularly the first 1000 days of life, is considered a critical window for immune development. The perturbations that occur when appropriate microbial signals are not received during this period may have long-lasting effects on the immune system resulting in susceptibility to allergic diseases. In this context, existing evidence suggests that breast milk composition has potential for preventing allergic diseases in early life, although protection afforded by breastfeeding against allergy development has been subject of controversy in the last decades. Despite some high-quality research, there is conflicting evidence on the protective role of breastfeeding in relation to many non-communicable diseases, including immunological outcomes. It has been hypothesized that conflicting results may be due to variations in human milk composition, as it is known to contain different concentrations of a wide variety of active immune components [34]. Some studies consider breastfeeding an important protective factor for the development of atopy [35], while others do not consider breastfeeding as a protective factor, or even predict a negative role in the increase in risk [36]. Reasons for this debate include methodological differences and bias in the research performed to date, the immunological complexity of breast milk composition [6], and possibly also genetic differences between patients. All of these factors may influence whether breastfeeding is protective against allergy development, or if it is sensitizing. Moreover, maternal diet can modify the immunological properties in breast milk, playing a role in childhood allergies as well [37]. As it is not possible to randomize breastfeeding exposure, evidence comes only from observational studies and this is the reason as to why it remains difficult to determine conclusive considerations. However, evidence suggests that there would be much to lose by not recommending breastfeeding; this is the reason that World Health Organization (WHO) and European Society for Pediatric Gastroenterology Hepatology and Nutrition (ESPGHAN) recommend exclusive breastfeeding for 4 to 6 months, reiterating that breast milk can be considered the most important preventive measure for allergic disease development in high-risk patients. Following these considerations, Guilbert et al. [38] showed that the respiratory function of adolescents fed with formula within the first 2 months of life was significantly reduced when compared to controls, exclusively breastfed for more than 4 months. In accordance with these data, a recent systematic review and meta-analysis by Lodge et al. [39] found evidence that breastfeeding reduces the risk of asthma in childhood, but they found weak evidence for reducing risk of AD up to 2 years and allergic rhinitis up to 5 years of age. They found no risk or protective association for food allergy.

In a mouse model, Verhasselt et al. [40] showed that the induction of tolerance can be mediated by oral intake of antigenic molecules, and a recent review by Vieira Borba et al. [41] highlighted that breastfeeding was associated with a protective role in type 1 diabetes and celiac disease, while benefits in type 2 diabetes were related to the prevention of obesity and metabolic syndrome later in life. This has led to the hypothesis that specific human milk molecules, such as intact human insulin, gliadin, and other food allergens (i.e., peanut proteins, ovalbumin, wheat, β-lactoglobulin, casein, and bovine γ-globulin) may be involved in the prevention of food allergy and autoimmune diseases [42]. Because they are molecules involved in the pathogenesis of immune-mediated diseases, their early intake can lead to a protective effect against early infections, the induction of antigen-specific tolerance and a better regulation of the infant’s microbiome. The authors of these studies evaluated the hypothesis that maternal exposure to certain antigens during lactation, the transfer of antigens into breast milk, and the presence of TGF-beta in breast milk were necessary for the induction of antigen tolerance in breast-fed infants and therefore for protection against allergic disease development. This observation is supported by studies in rodents, showing that the transfer of antigens through breast milk prevents the induction of the antigen-specific immune response in the newborn and the development of allergic diseases later in life [43,44]. In this context, it is worth mentioning the 2014 Cochrane meta-analysis by Kramer et al. [45], which demonstrates that antigen avoidance diet during pregnancy, and also during breastfeeding, does not reduce the possibility of allergy occurrence to cow’s milk, egg, and peanuts in the first, second, and seventh year of life. This is in accordance with US guidelines [46] and European recommendations [47] which do not recommend observance of eliminating diets in pregnant women as an effective measure for allergy prevention in children. Only in breast-fed infants suffering direct symptoms due to maternal intake of food allergens should the mother eliminate the offending foods and follow a dietetic regimen.

Other studies show an inverse relationship between the duration of breastfeeding and the prolongation of wheezing in early childhood. Kull’s study showed that breastfeeding for less than 4 months, and the early introduction of complementary food after only 3 to 4 months of exclusive breastfeeding, increased the risk of asthma at the age of 4 [48], while Elliot showed similar protection against wheezing for the first 3 years, but not at 7 to 8 years [49]. However, not all studies demonstrate these protective effects. Recent studies have raised concern that exclusive breastfeeding may not protect children and may even increase the risk of the development of allergic conditions, such as AD. Giwercman et al. [36] found that longer breastfed infants had an increased risk of AD development, while the risk of wheezy disorder and severe wheezy exacerbations was reduced. Evidence on allergic rhinitis (AR) is limited, although in 2002 Bloch et al. [50] performed a meta-analysis of prospective studies to evaluate the effects of exclusive breastfeeding on the development of AR, finding that exclusive breastfeeding for at least 3 months of life had a protective role against the development of AR. More recently, Lodge et al. [39] evaluated the association between breastfeeding and AR showing that a reduced risk of AR in patients under 5 years of age was associated with breastfeeding, while there was no association after 5 years of age. Again, however, data are limited because, although the authors of this meta-analysis supported the benefit of breastfeeding in the prevention of AR, they acknowledge that the protective effects of breastfeeding may have been confused by the well-known protective effect of human milk against viral respiratory infections. Indeed, given the difficulty of differentiating between AR and viral rhinitis in young children, Lodge et al. hypothesize that a reduction in viral respiratory infections has possibly been interpreted as a reduction in rhinitis symptoms.

Moreover, continued breastfeeding while solids are introduced into the diet and delaying the introduction of solids until at least 17 weeks of age are associated with fewer food allergies, as showed in the clinical cohort trial by Grimshaw et al. [51]. This study supports the current American Academy of Pediatrics allergy prevention recommendations, and the European Society of Pediatric Gastroenterology, Hepatology and Nutrition’s recommendations on complementary feeding regarding not introducing solids before 4 to 6 months of age. It also supports the American Academy of Pediatrics breastfeeding recommendations that breastfeeding should continue while solids are introduced into the infant diet.

### 3.1. Asthma

In 2001, a study by Gdalevich and Mimouni [52] revealed an association between breastfeeding and a reduced prevalence of asthma in children, which was subsequently confirmed in a recent meta-analysis by Dogaru et al. [53]. Some of the protective effects described may be mediated by strengthening or maturation of the child’s immune system. However, there is a wide heterogeneity between studies reporting an inverse association between breastfed infants and asthma development. Some of the differences can be explained by variations in the definitions of exclusivity and duration of breastfeeding, and by the methods for diagnosing asthma in children. Indeed, it is now well known that many children who suffer from wheezing early in life do not develop asthma in school-age or adolescence, yet wheezing is often used as a diagnostic indicator of asthma [48]. It is worth noting that breastfeeding’s protective effects on asthma are more evident in recent studies and probably better quoted in the methodology [53]. The subgroup analysis shows a greater protective effect of breast milk in developing countries where allergic conditions are less common; thus it seems likely that the primary protective effect of breast milk is on viral wheezing rather than atopic asthma [39,54].

### 3.2. Atopic Dermatitis (AD)

Most of the data regarding the association between breastfeeding and the development of AD come from birth cohorts or cross-sectional studies. In a large observational study that evaluated more than two hundred thousand babies worldwide, The International Study of Asthma and Allergies in Childhood (ISAAC), the authors could not find evidence for a protective effect of breastfeeding on the development of AD at 6 to 7 years of age, although they reported protection against its severe form [55]. The results of the systematic review and meta-analysis by Lodge et al. [39] show that infants exclusively breastfed for more than 3 to 4 months have a lower risk of developing AD, although this protective effect ceases after 2 years. The authors also highlighted a high risk of bias from smaller studies showing more significant protective effects.

### 3.3. Food Allergy

Studies evaluating the association between breastfeeding and food allergy have yielded mixed results. Some cohort studies report a reduced risk of developing food allergies in both a general population [48,56] and high-risk infants, such as preterm infants [57], while other research suggests an increased risk with breastfeeding [58]. One of the most recent systematic reviews and meta-analysis found no statistically significant association between breastfeeding and the development of food allergies [39]. Assessing the risk of food allergy is not easy in the context of a clinical trial, as the gold standard for confirming the diagnosis is the double-blind food challenge, which is not always a viable option for study participants. In many studies, a combination of clinical history and skin prick tests or serum IgE tests is used as a surrogate marker for a diagnosis of food allergy with inevitable high heterogeneity. Therefore, the main target of future research should be the harmonization of results definition [39]. Recent clinical studies showing benefits in the early introduction of food (from 3 to 4 months of age) while breastfeeding, may indicate a useful strategy to reduce the risk of developing food allergies. This has been driven by recent studies, such as the Learning Early About Peanut Allergy (LEAP) [59] and Enquiring About Tolerance (EAT) [60], suggesting that in some infants, introduction of peanuts and egg proteins before 6 months of age reduce the risk of allergy to these foods [61].

## 4. Conclusions

In the past few decades, an increase in the prevalence of asthma and other allergic conditions, such as atopic dermatitis and food allergies, was observed. Since 1936, when Grulee and Sanford [62] reported a significantly lower incidence of atopic dermatitis in breast-fed infants compared with formula-fed infants, scientific attention has been placed on the protective role of breastfeeding in the development of allergies. However, many further studies have not been able to provide the definitive answer as to whether or not the development of allergic diseases can be prevented by breastfeeding. It has long been known that breastfeeding provides health benefits, but it is not understood whether, or how, breastfeeding decreases the risk of developing allergies. Evaluating breastfeeding’s potential to prevent allergic disease in observational studies is not an easy challenge as a wide range of factors, such as socioeconomic status, positive allergy family history, early exposure to pets and smoking, and timing of weaning, alongside variations in breast milk composition, are all sources of bias. What is certain is that human milk remains the gold standard nourishment for the newborn up to 6 months of life, as it is widely demonstrated that breastfeeding leads to positive health outcomes by protecting against many infectious and chronic diseases. There remains no doubt that breastfeeding is a major investment in health for the offspring, for the mother, and for the whole community. For these reasons, WHO and ESPGHAN recommend starting breastfeeding immediately after birth with skin-to-skin contact, promoting exclusive breastfeeding for the first 6 months of life and, once weaning has begun, to continue breastfeeding up to 2 years of age and over. Solid foods should not be introduced before 4 to 6 months of age and breastfeeding should continue while solids are introduced into the diet for 1 year, or longer, as mutually desired by mother and infant.

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
