# Peer review of "Breastfeeding and Allergic Diseases: What’s New?"

_children, 2021, doi:10.3390/children8050330_

Round 1

Reviewer 1 Report

The manuscript “Breastfeeding and allergic diseases: what,s new?” has been re-reviewed. According to the reviewer the corrections introduced by the authors (doubtful sentences removed, inserted relevant information regarding current guidelines on women,s diet during pregnancy and breast feeding, as possible protection against allergy, insertion of new relevant references and correction of citations of the references) significantly improved its quality.

Currently, the reviewer suggests the following changes:

  • Line 113: oligosaccharides in human milk (human milk oligosaccharides, HMO) are non-nutritive to the infant; oligosaccharides mentioned in line 113 are the same oligosaccharides previously mentioned (in line 94)
  • Line 128: secretory immunoglobulin A (sIgA)
  • The text fragment (lines 191-201) is hard to understand and should be restructured or subjected to correction by a native speaker
  • For the benefit of readers, the next paragraph (from line 202) should be completed with practical conclusion on complementary feeding from clinical cohort trial by Grimshaw et al [Grimshaw KE et al.: Introduction of complementary foods and the relationship to food allergy. Pediatrics 2013; 132(6): e1529-38] “ for reduce food allergy risk not introduce solids before 4 to 6 months of age as well as breastfeeding continue while solids are introduced into the diet and breastfeeding continue for 1 year, or longer, as mutually desired by mother and infant. The results of this study support the current American Academy of Pediatrics' allergy prevention recommendations, American Academy of Pediatrics' breastfeeding recommendations and the European Society of Pediatric Gastroenterology, Hepatology and Nutrition recommendations”. This conclusion can be possibly added to the Conclusion section
  • Please check the citations; the reviewer did not find references #42 and #50 in the text
  • Technical notes to References: should be?:
  1. Eidelman, A.I. Breastfeeding and the use of human milk: an analysis of the American Academy of Pediatrics 2012 Breastfeeding Policy Statement. Breastfeeding medicine 2012, 7, 323-324.
  2. Organization, W.H. WHO recommendations on postnatal care of the mother and newborn; World Health Organization: 2014.
  3. Dewey, K.G.; Heinig, M.J.; Nommsen, L.A. Maternal weight-loss patterns during prolonged lactation. The American journal of clinical nutrition 1993, 58, 162-166.

And similar below (lower case in the journal titles).

Reviewer 2 Report

Dear Author,

With great interest, I have read your paper on breastfeeding and allergic diseases. I would recommend a few clarifications in the paper, though, before its acceptance for publication. Please find my comments below:

  1. The evidence on long-term benefits, such as, e.g., improved cognitive performance, is limited. Especially in high-resource countries, breastfeeding is associated with higher socioeconomic status and the mother's educational status, less smoking, and obesity. Those factors by themselves might influence the child's development, a benefit that might be attributed to breastfeeding with no hard evidence of causation of this intervention. Therefore I would recommend including those limitations of the evidence into the passage on breastfeeding benefits.
  2. In the passage on asthma, I  propose to include the "Retrospective cohort study of Breastfeeding and the risk od childhood asthma" Lossius AK, Magnus MC at al. J Pedr 2018, a prospective cohort nationwide study.
  3. It might be worth considering adding a passage on allergic rhinitis.
  4. There might be some misconception in the reference section where you cite LEAP study results with ref No 29 from 2001, while the LEAP study groundbreaking results were released in NEJM in 2015. Please check the correctness of the references.

I am looking forward to reviewing the evised version of the manuscript.

Best regards
